# Hydrogen as a Temporary Alloying Element for Establishing Specific Microstructural Gradients in Ti-6Al-4V

**Christopher David Schmidt \*** , **Hans-Jürgen Christ and Axel Von Hehl**

Institut für Werkstofftechnik, Universität Siegen, Paul-Bonatz-Straße 9-11, 57076 Siegen, Germany;
hans-juergen.christ@uni-siegen.de (H.-J.C.); axel.vhehl@uni-siegen.de (A.V.H.)
\* Correspondence: christopher.schmidt@uni-siegen.de

**Abstract:** Parts of vehicles, such as landing gear components of aircrafts, are subject to growing demands in terms of sustainability via lightweight design and durability. To fulfill these requirements, the development of thermochemical processes is auspicious. Titanium alloys allow a heat treatment in hydrogen-containing atmosphere for temporary hydrogen alloying, often called thermohydrogen treatment (THT). The investigation presented intends to realize a local microstructure modification of Ti-6Al-4V by means of THT. The study aims to use hydrogen (H) as a promoter for changing the local distribution and morphology of strengthening precipitates during THT as well as the local grain size (microstructural gradient). Both shall improve the fatigue properties of the material after hydrogen degassing. To derive suitable thermohydrogen treatment process parameters, the resulting fatigue crack propagation resistance and fracture toughness after different solution heat treatments are determined experimentally and compared to each other. Moreover, various graded microstructures are evaluated after hydrogen uptake (hydrogenation) and hydrogen degassing (dehydrogenation) using numerically simulated hydrogen concentration profiles, observed hardness curves, metallographically determined microstructure gradients and the corresponding results of the phase analysis by means of X-ray diffraction. The study shows that hydrogenation at 500 °C and dehydrogenation at 750 °C enables the generation of a promising microstructural gradient.

**Keywords:** thermohydrogen treatment; numerical simulation of hydrogen concentration profiles

## 1. Introduction

Titanium alloys are classified according to the β (bcc) phase stability into the alloy classes α(hex), (α + β) and β titanium (Ti) alloys [1]. Due to the high gas solubility of the β phase and the complete reversibility of the metal–gas reaction, temporary alloying with atomic hydrogen (H) is possible as part of a thermal treatment, a so-called thermohydrogen treatment (THT) [2]. THT usually consists of the process sequence solution treatment (ST), diffusion-controlled hydrogen uptake (hydrogenation), hydrogen degassing (dehydrogenation) and aging. With few exceptions [3] it is applied to Ti alloys only, aiming for homogeneous microstructure adaptation [4–12] and the generation of microstructural gradient [13,14]. A schematic visualization of the process, embedded into the Ti-6Al-4V/hydrogen phase diagram is shown in Figure 1. The steps of solution treatment (step 1) and aging (step 4) are carried out in ambient atmosphere (red scripture in Figure 1) at temperatures A and D. Hydrogenation is executed in a gas mixture containing hydrogen (step 2, blue scripture in Figure 1) at temperature B, and dehydrogenation takes place in vacuum (step 3, black label in Figure 1) at temperature C. The figure shows two different approaches depending on the H content in the near-surface area of the sample that is established after the hydrogenation. The study follows two different concepts of THT (path A, red dotted lines and Bm blue dotted lines), which differ in terms of maximal hydrogen concentration established in the near-surface area.

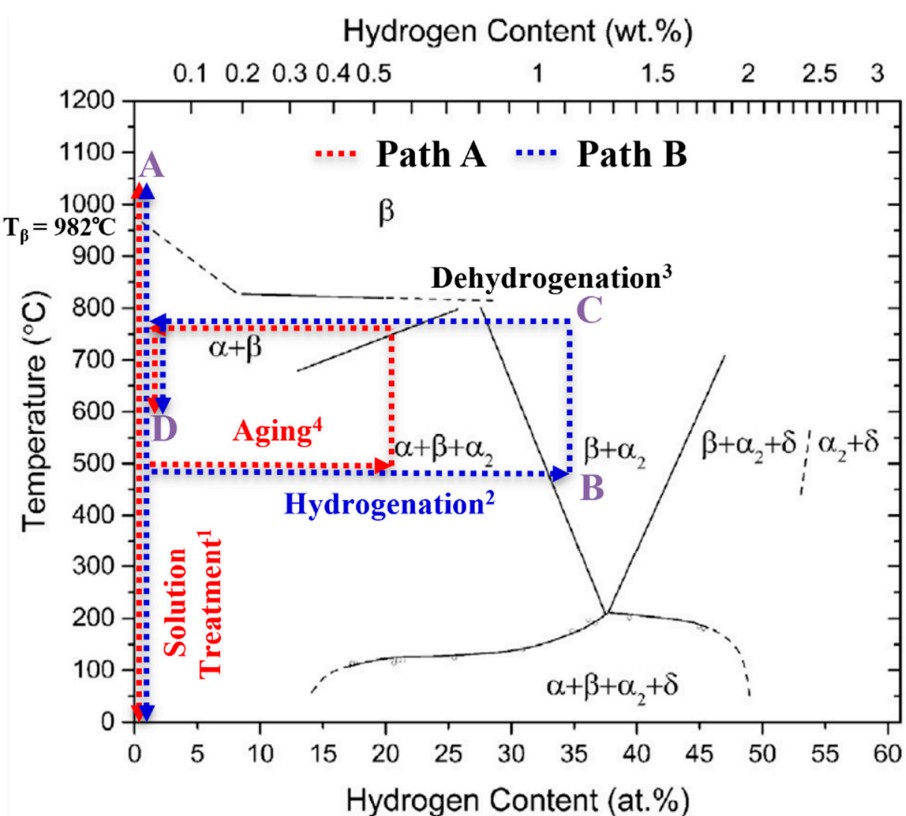

**Figure 1.** Schematic representation of the THT process steps correlated with the Ti-6Al-4V/H phase diagram, (reprinted/adapted with permission from [4]. 2014, Acta Materialia Inc.), and indicating the locally occurring maximal hydrogen concentration; the superscripts correspond to the usual process step sequence.

Hydrogen-induced effects that influence the microstructure (extrinsic H effects) can have a positive impact on the mechanical properties of the material. The extrinsic H effects can be described by two main phenomena: Firstly, hydrogen is a strong β-stabilizing element in Ti alloys and lowers the β-transus temperature $T_β$ to $T_β$ (H) (see Figure 1). Hence, the temperature of solution treatment can be lowered, reducing grain growth during the process as compared to conventional heat treatment (solution treatment and aging) of Ti alloys, if hydrogenation is applied before the ST. Secondly, when the hydrogenation concentration in a sample exceeds the maximum hydrogen solubility, hydrogen evokes hydride formation, which is associated with local volume expansion. The hydride-induced volume expansion in the α and β phase reaches a value between 17% and 25%, depending on the underlying solid solution (chemical composition) [5]. After the hydrides are dissolved during dehydrogenation, dislocations and vacancies remain, which act as additional nucleation sites for precipitates. This phenomenon enhances the α precipitation (and thus a more homogeneous and finer precipitate morphology) as well as the recrystallization kinetics (and thus a potentially reduced grain size) during a subsequent heat treatment [6]. Previous work on THT has led to increased strength under cyclic and static loading by producing a homogeneous and, compared to conventionally heat-treated Ti alloys, finer microstructure [7,8].

The aim of the research project presented is the realization of a local microstructure which, depending on the surface distance, provides a microstructure gradient that improves the properties relevant to technical applications like the strength of components of an aircraft at cyclic loading in HCF and LCF regime. Compared to a homogeneous microstructure, a fine, equiaxed microstructure in the near-surface region is to be established using THT (process comparable to [9]), which significantly prolongs the fatigue crack initiation phase, while a coarse, lamellar microstructure is to be achieved in the interior,

which slows down the propagation of long fatigue cracks [14,15]. Berg and Wagner [16] were able to evoke fatigue life increasing microstructure gradients in Ti alloys using thermomechanical treatments. In contrast to conventional processes enabling surface-near microstructural modifications, such as shot peening or other thermomechanical treatments, THT uses hydrogen from the gas phase as a promoter for the intended local microstructural modifications. Hence, THT enables a local microstructure adaptation of complex geometries, like tubes with a variable wall thickness, that cannot be surface hardened via thermomechanical surface treatments. Therefore, THT is applicable to components which possess a high geometrical complexity and promises an extension of the degrees of freedom for property improvement of applications. The study presented compares the fatigue crack propagation resistance and the fracture toughness in dependence of three different solution treatment conditions. The comparison is used to select suitable solution treatment parameters. The solution treatment must evoke a morphology that is suitable for the surface-far area of the final microstructural gradient [16]. For the further selection of the hydrogenation and dehydrogenation parameters, different graded microstructural states after hydrogenation and dehydrogenation are evaluated by means of simulated hydrogen concentration ($c_H$) profiles, measured hardness curves, metallographically recorded microstructural gradients and XRD analysis of the resulting phase fractions.

## 2. Materials and Methods

The widely used ($\alpha + \beta$)-Ti alloy Ti 6Al-4V was used as the test material. For the selection of suitable solution treatment parameters, the solution treatment conditions after [15] were assessed with respect to the threshold stress intensity range for long cracks ($\Delta K_0$) and the fracture toughness ($K_{Ic}$ or $K_Q$). $\Delta K_0$ was determined with four-point bending fatigue tests at 100 Hz (stress ratio: 0.1) by using a 4-point bending resonance and a crack length measuring device (Russenberger Prüfmaschinen AG, Neuhausen am Rheinfall, Switzerland) and the load-shedding method, in which the stress amplitude is reduced according to an exponentially decreasing function. $\Delta K_0$ is achieved when the crack propagation rate $da/dN$ is less than $10^{-11}$ m/load cycle. For the determination of fracture toughness, compact tension tests were carried out at a servo-hydraulic testing machine (MTS Systems Corporation, Eden Prairie, MN, USA) using a strain gauge. The specimen geometries were designed according to the guidelines of ASTM E 647 ($\Delta K_0$) and ASTM E 399 ($K_{Ic}$ or $K_Q$).

To determine the (de)hydrogenation parameters, (de)hydrogenation of cylindrical samples of 5 mm in diameter and 100 mm in length were heat-treated in a horizontal (vacuum) furnace (University of Siegen, Siegen, Germany) under supply of a He/$H_2$ gas mixture with 10% $H_2$ (hydrogenation) or in vacuum ($p = 2 \cdot 10^{-5}$ mbar, dehydrogenation). The sample surfaces were prepared according to [16], including a Pd coating, that enables hydrogen uptake at comparably low temperatures (500 °C and 600 °C). Moreover, the samples were placed on a Zr foil that acts as a getter of gas impurities, particularly oxygen. The analysis of the hydrogen concentration was conducted by means of carrier hot gas extraction (Leco Corporation, St. Jospeh, MI, USA). The hydrogenation experiments were intended to induce $c_H$ gradients that would promise the desired microstructure after dehydrogenation. Therefore, two approaches (path A and B) were pursued, which should differ regarding the maximum hydrogen concentration generated in the near-surface area of the sample (see Figure 1). The realization of the different hydrogen concentration values is done by applying different hydrogen partial pressures (path A: 20 mbar, deeper penetration depth; path B: 100 mbar: steeper gradient of microstructural changes).

For the design of path A, samples of the three different initial microstructures, generated by different solution treatments, were hydrogenated under a hydrogen partial pressure ($p_{H2}$) of 100 mbar at 500 °C and 600 °C at 2, 3, 4, and 6 h (acc. to [15]). The parameters for the different solution treatment conditions are listed in Table 1.

**Table 1.** Parameter for solution annealing and resulting microstructural parameters.

| Solution Treatment | A (Lamellar) | B (Bimodal) | C (Equiaxed) |
|---|---|---|---|
| Temperature and duration | 1050 °C at 1 h | 982 °C at 1 h | 950 °C at 1 h (I) + 720 °C at 1 h (II) |
| Quenchant | furnace atmosphere | water | I: furnace atmosphere, II: air |
| β phase fraction (vol.%) | 14.1 | 18.5 | 15.8 |
| α particle size (μm) | 18.3 | 5.6 | 16.3 |

To determine the hydrogenation temperature ($T_H$) and duration ($t_H$), the hydrogenation conditions were selected, which led to the maximum possible $c_H$–value in the near-surface area of the sample without causing surface cracks longer than 100 μm. After the hydrogenation treatment, the samples were water quenched in order to stop a further hydrogen diffusion and release. This should enable a more precise determination of the amount of absorbed hydrogen.

The design of path B is based on the results of Berg and Wagner [16]. They reported that the near-surface region of the graded microstructure leads to maximum improvement in fatigue strength at a penetration depth of about 750 μm. Hence, hydrogenation experiments were carried out by varying the pressure $p_{H2}$ (20, 40, 60, 80 and 100 mbar) to produce a $c_H$ profile in which the local $c_H$ reaches hydrogen concentration values sufficient for hydride formation (>15 at.%) only in a region from the surface to a distance from the surface of 750 μm. The evaluation of the influence of the different $p_{H2}$ on the resulting microstructure was done with respect to the maximum possible $p_{H2}$ that produced no or negligible surface cracks.

To determine the (de)hydrogenation temperature ($T_H$, $T_D$) and duration ($t_H$, $t_D$) (de), hydrogenated samples of the three solution treatment conditions dealt with (see Table 1) were examined by means of SEM (FEI Company, Hillsboro, OR, USA) (used for a qualitative evaluation of the stereological parameters). Accordingly, the referring $c_H$ profiles were numerically simulated using Matlab (R2020, The MathWorks, Natick, MA, USA). The numerical simulation of the $c_H$ profiles consider the $T_H$- and $p_{H2}$-dependent incubation time determined in [15] and deliver a kinetic surface correction factor (SCF) of the hydrogen adsorption and desorption process, which is the ratio of the experimental and the numerically calculated hydrogenation time. The resulting $c_H$ profiles, the hydrogenation and dehydrogenation times as well as the surface correction factors were calculated according to the method described in [15] by using finite-element method (FEM) and Matlab (R2020). The input parameters for the simulation consist of thermodynamic data (hydrogen solubility) and kinetic data (hydrogen diffusion coefficient) and are reported in [15] as well and are displayed in Tables 2 and 3.

**Table 2.** Hydrogen diffusion coefficient for different solution treatment conditions, data from [15].

| Temperature (°C) | Hydrogen Diffusion Coefficient $D_H$ (m²/s) | | |
|---|---|---|---|
| | Lamellar | Bimodal | Equiaxed |
| 200 | $4.0 \times 10^{-10}$ | $5.0 \times 10^{-11}$ | $1.1 \times 10^{-10}$ |
| 300 | $1.5 \times 10^{-10}$ | $9.0 \times 10^{-11}$ | $2.0 \times 10^{-10}$ |
| 400 | $4.0 \times 10^{-10}$ | $4.0 \times 10^{-10}$ | $3.0 \times 10^{-10}$ |
| 500 | $9.0 \times 10^{-10}$ | $1.0 \times 10^{-9}$ | $5.5 \times 10^{-10}$ |
| 600 | $2.8 \times 10^{-9}$ | $2.0 \times 10^{-9}$ | $2.1 \times 10^{-9}$ |
| 700 | $6.0 \times 10^{-9}$ | $4.0 \times 10^{-9}$ | $4.5 \times 10^{-9}$ |
| 800 | $8.5 \times 10^{-9}$ | $6.0 \times 10^{-9}$ | $5.0 \times 10^{-9}$ |

**Table 3.** Hydrogen saturation concentration coefficient at 500 °C and 600 °C for different solution treatment conditions, data from [15].

| $p_H$ (mbar) | H Saturation Concentration (at.%) | | | | | |
|---|---|---|---|---|---|---|
| | 500 °C | | | 600 °C | | |
| | Lamellar | Bimodal | Equiaxed | Lamellar | Bimodal | Equiaxed |
| 1 | 4.8 | 3.1 | 2.8 | 2.50 | 2 | 2.3 |
| 4 | 10.3 | 8.2 | 6.7 | 3.80 | 3.2 | 2.7 |
| 9 | 14.6 | 11.0 | 8.5 | 4.75 | 3.6 | 4.0 |
| 20 | 19.2 | 12.4 | 12.7 | 5.31 | 6.3 | 6.4 |
| 40 | 24.8 | 16.6 | 22.0 | 10.60 | 9.2 | 13,0 |
| 60 | 33.3 | 18.8 | 30.8 | 13.83 | 13.8 | 15.8 |
| 80 | 34.3 | 19.4 | 33.4 | 18.61 | 20.3 | 20.1 |
| 100 | 36.1 | 26.6 | 40.8 | 21.68 | 26.4 | 26.4 |

The hydrogenation time for path B was calculated by FEM using the surface correction factor of the hydrogenation from path A to estimate the necessary hydrogenation time. Furthermore, local Vickers hardness profiles were determined applying hardness testing (Strues GmbH, Willich, Germany) (2 kp for 10 s). Moreover, the stereological parameters were measured on SEM-BSE micrographs via image processing (Lince 2.4.2 ß, TU Darmstadt, Darmstadt, Germany and ImageJ 1.52a, National Institutes of Health, Bethesda (MD), USA), and X-ray diffraction (XRD) (Malvern Instruments, Malvern, United Kingdom) phase analysis was performed at 45 kV and 40 mA.

## 3. Results and Discussion

### 3.1. Specifying the THT Parameter Values

Table 4 lists the results obtained for $\Delta K_0$ and $K_Q$ in dependance of the solution treatment condition. Since the criteria defined by ASTM E 399 for determining the fracture toughness under plane load ($K_{Ic}$) were not completely fulfilled, the conditional fracture toughness ($K_Q$) is used in the following. This is equally suitable for comparing the fracture toughness of different solution treatment conditions.

**Table 4.** $\Delta K_0$ and $K_Q$ for different solution treatment conditions.

| Solution Treatment | $\Delta K_0$ (MPa$\sqrt{m}$) | $K_Q$ (Mpa$\sqrt{m}$) |
|---|---|---|
| A | 4.9 | 94 |
| B | 15.5 | 61 |
| C | 5.0 | 61 |

Comparing the solution treatment conditions, the bimodal microstructure (resulting from solution treatment B) exhibits the highest stress intensity range for long fatigue cracks $\Delta K_0$, while the lamellar microstructure (resulting from solution treatment A) provides the highest fracture toughness $K_Q$ (Table 4). The equiaxed microstructure (resulting from solution treatment C) shows inferior resistance against crack propagation under cyclic and monotonic loading and was therefore excluded as a suitable initial condition for THT. The lamellar microstructure (resulting from solution treatment A) shows by far the highest value of $K_Q$ [1]. However, since the results of $\Delta K_0$ promote the bimodal microstructure (resulting from solution treatment B), the susceptibility to crack formation after hydrogenation (Figure 2) was additionally considered as a criterion of selection. The different behavior in terms of crack initiation is depicted exemplarily for the lamellar (after solution treatment A) and the bimodal (after solution treatment B) microstructure after hydrogenation at 500 °C for 2 h.

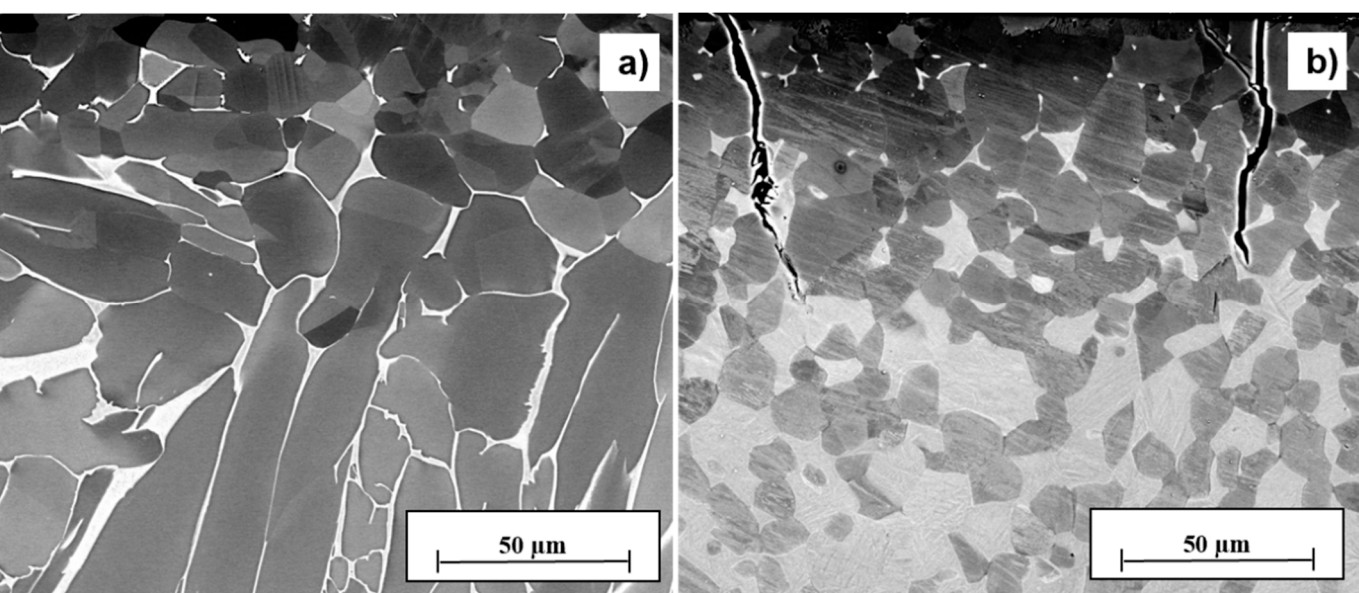

**Figure 2.** Exemplary micrographs of the near-surface area of Ti-6Al-4V samples after solution treatment A (**a**) and B (**b**), hydrogenated at 500 °C for 2 h.

The dark gray areas in Figure 2 correspond to the $\alpha$ phase, while the white areas represent the $\beta$ phase. The light gray areas in Figure 2b show the fine Widmanstätten or basket-weave microstructure ($\alpha$ colony) of the bimodal Ti microstructure. In comparison to the center, the near-surface area exhibits a higher cooling rate after solution treatment. In combination with absorption of oxygen at the surface, this may have caused the generation of equiaxed particles found at 50 μm distance from the surface after the solution treatment in both solution-treated conditions. The hydrogenation of the lamellar microstructure (solution treatment A) at $T_H$ = 500 °C and $t_h$ = 2 h (Figure 2a, $c_H$ = 12 at.%) delivers a crack-free sample, while the hydrogenation of the bimodal microstructure after solution treatment B at the same temperature, aiming for a comparable hydrogen concentration (10 at.%), shows massive cracks in the near-surface area (Figure 2b). The cracks may be a consequence of the hydride phase precipitation or the $\alpha_2$ phase formation that may occur after hydrogenation [4]. This assumption is discussed in the following by means of the local hydrogen concentration ($c_H$ profiles, Figure 3) and the XRD phase analysis. Since the unit cell of the hydride phase has a 17–25% larger specific volume than the unit cell of the $\alpha$ and $\beta$ phase, a certain hydride phase volume fraction can cause local cracks. The lamellar morphology (solution treatment A) is the only one that shows no cracks at a hydrogen concentration of more than 10 at.%. Therefore, this morphology seems to tend less to hydride-induced cracks than the other microstructures tested. This observation might be due to the superior fracture toughness (see Table 4) and ductility [1] of the lamellar microstructure (solution treatment A) as compared to the bimodal (solution treatment B) and equiaxed microstructure (solution treatment C). The 600 °C hydrogenation experiments of the three solution treatment conditions evoked microcracks or cracks in all cases, even at smaller averaged hydrogen concentration reached (e.g., 6 at.%). Since the hydrogen solubility at 600 °C is less than at 500 °C [15] and therefore hydride-induced cracking of the maximum hydrogenated surface area should be more likely at 500 °C, this observation was not expected. The reason for that could be an increased $\alpha_2$ phase fraction after hydrogenation at 600 °C, which is a brittle phase and can cause cracks at the surface at water quenching. In summary, the crack observation results (Figure 2) in combination with the results of the determination of crack propagation resistance and fracture toughness (Table 4) lead to the conclusion that solution treatment A (lamellar morphology) and a hydrogenation temperature of 500 °C are suitable parameters for the THT paths A and B. Hence, these parameters were selected for the further experiments of the study presented.

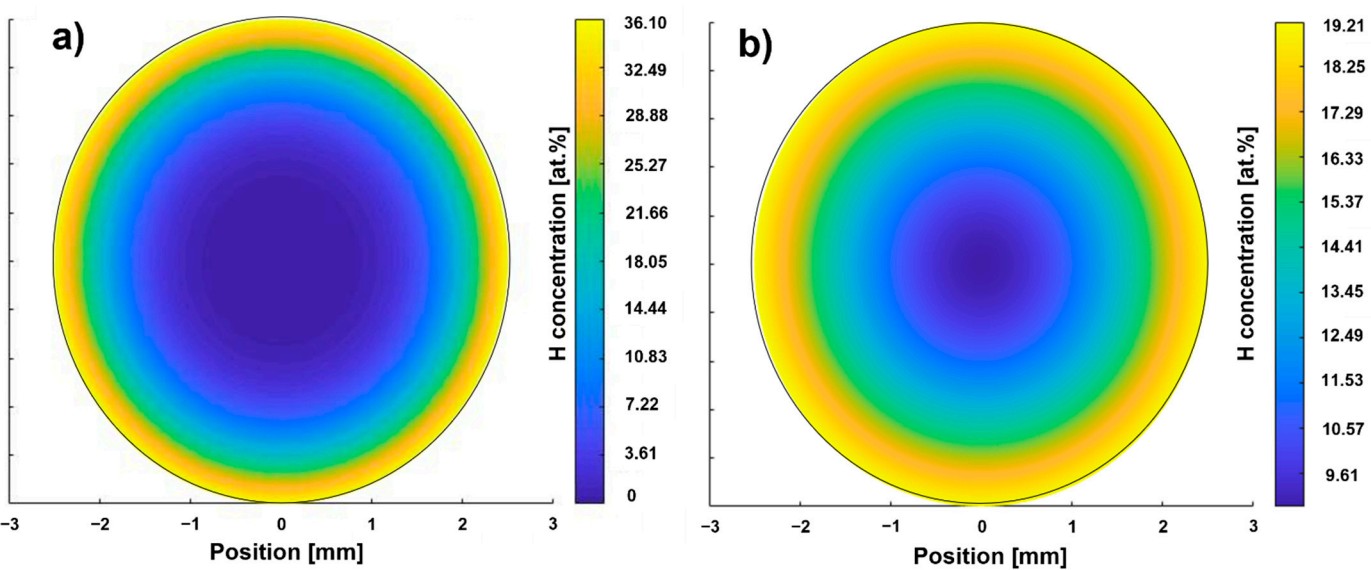

**Figure 3.** Calculated hydrogen concentration profiles according to path A (**a**) and B (**b**).

The specification of the dehydrogenation temperature ($T_D$) for path A and B is based on the results of Dunstan et al. [8], in which tensile specimens were hydrogenated homogeneously up to a hydrogen concentration comparable to the one used at the surface in the present investigation (35 at.% in [8] as compared to 36 at.% in this study). Dehydrogenation was carried out in [8] at temperatures in the regime between 600 °C and 950 °C. The results of tensile tests displayed the best compromise between strength and ductility for $T_D = 750$ °C and therefore was used for the study presented.

*3.2. Evaluation of Microstructural Gradients*

Table 5 lists the THT parameters used for both paths, the resulting experimentally determined hydrogen concentration averaged over the cross section ($c_H$), the resulting calculated surface correction factor (SCF) for hydrogenation ($SCF_H$) and dehydrogenation ($SCF_D$) and the incubation time ($t_i$, determined in [15]) used for the FEM calculations.

**Table 5.** Overview of the parameters used and calculated for THT.

| Path | Hydrogenation | | | | | | Dehydrogenation | | | | |
|---|---|---|---|---|---|---|---|---|---|---|---|
| | $p_H$ (mbar) | $T_H$ (°C) | $t_H$ (h) | $c_H$ (at.%) | $t_i$ (s) | $SCF_H$ | $T_D$ (°C) | $t_D$ (h) | $c_H$ (at.%) | $t_i$ (s) | $SCF_D$ |
| A | 100 | 500 | 2 | 11.65 | 160 | 14.35 | 750 | 7 | 0.16 | 220 | 17.73 |
| B | 20 | | 6.15 | 15.32 | 230 | 12.43 | | 6 | 0.15 | | 14.25 |

The hydrogen concentration of the path B sample is higher than that of the path A sample (Table 5). The hydrogen concentration after dehydrogenation corresponds to the hydrogen concentration of the as-received condition. The surface correction factor for hydrogenation according to path B is lower than that of path A, which was used for the calculation of the hydrogenation duration for path B. Therefore, a penetration depth slightly higher than the target value of 750 µm is to be expected for path B. The dehydrogenation surface correction factors for path A and B are of the same magnitude. The $c_H$ profiles after hydrogenation of the two paths predicted by FEM calculation is shown in Figure 3.

The hydrogen concentration profile according to path A (Figure 3a) shows a maximum local hydrogen concentration of 36.10 at.% at the surface. The depth at which the H content is sufficient to enable hydride precipitation (15 at.%) is referred to in the following as penetration depth and is less than 750 µm. The result shows that an area in the center with a diameter of 3 mm is between 0 and 7 at.%. The hydrogen concentration at the surface

of the sample of path B is lower (19.21 at.%, Figure 3b) than the one of path A, while the penetration depth is larger (approx. 1000 µm). The center of the path B sample exhibits half of the surface concentration. Hence, these calculation results are in reasonable agreement with what the hydrogenation parameter suggests: that THT path A was executed at a higher hydrogen partial pressure, resulting in a higher saturation concentration [15] and therefore enabling a higher hydrogen uptake at least in near-surface areas. The significantly longer hydrogenation duration of path B results in a larger penetration depth as well as a larger hydrogen concentration in the center. According to both hydrogen concentration profiles, the hydrogenation parameters of both paths seem to allow the adjustment of a hydrogen concentration gradient. Figure 4 compares the hardness profiles (HV2 versus distance to surface) of the two paths after hydrogenation and dehydrogenation with the hardness profile of the solution treatment condition (ST).

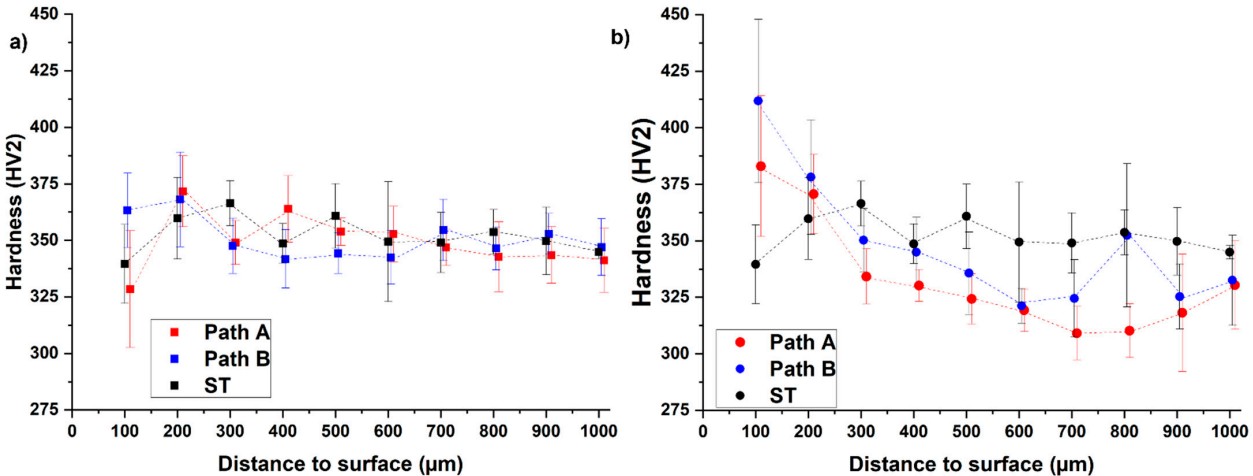

**Figure 4.** Measured hardness curves after hydrogenation (**a**) and dehydrogenation (**b**) for both paths in comparison to its initial condition (ST, solution treatment A).

The result of the hardness profiles shows that for the hydrogenated samples (blue and red data points), the hardness seems to decrease with increasing distance to surface (Figure 4a) except for a few data points adjacent to the surface. Nevertheless, the results do not prove a relationship between the hydrogen concentration and the hardness. Moreover, the hardness profiles after hydrogenation are comparable to the hardness profile of the initial condition (solution treatment, black data points). In the comparison of the two hydrogenated samples, it is noticeable that the hardness of the path A sample is above that of the path B sample in the range of 200–600 µm and below it thereafter (700–1000 µm).

The hardness curves after dehydrogenation show that the hardness decreases with increasing distance to surface in a range of 200 µm distance to surface, and the level of hardness of the path B sample is above the path A sample over the entire measuring range (Figure 4b). The dehydrogenated THT-samples show higher hardness values than the solution-treated sample at 100 µm distance to surface (path A: 13% increased, path B: 21% increased) and 200 µm distance to surface (path A: increase of 3%, path B: increase of 5%). At values of more than 200 µm distance to surface, the dehydrogenated samples show less hardness than the solution-treated reference, which might be since the additional heat treatment steps have reduced the strength of this inner microstructure area. At 100–300 µm the hardness values of the dehydrogenated samples are higher than the corresponding values of the hydrogenated samples. This observation supports the assumption that there is no proportional relationship between the local hydrogen concentration and the hardness. In general, the dehydrogenation of the hydrogenated condition enables the formation of the desired microstructural gradient, although the strength-enhanced area does not extend to 750 µm but 200–300 µm. This leads to the assumption that an H content of 15 at.% is not

sufficient to enable hydride precipitation. To verify this, Figure 5 compares the SEM-BSE images of the microstructures of both THT routes after dehydrogenation.

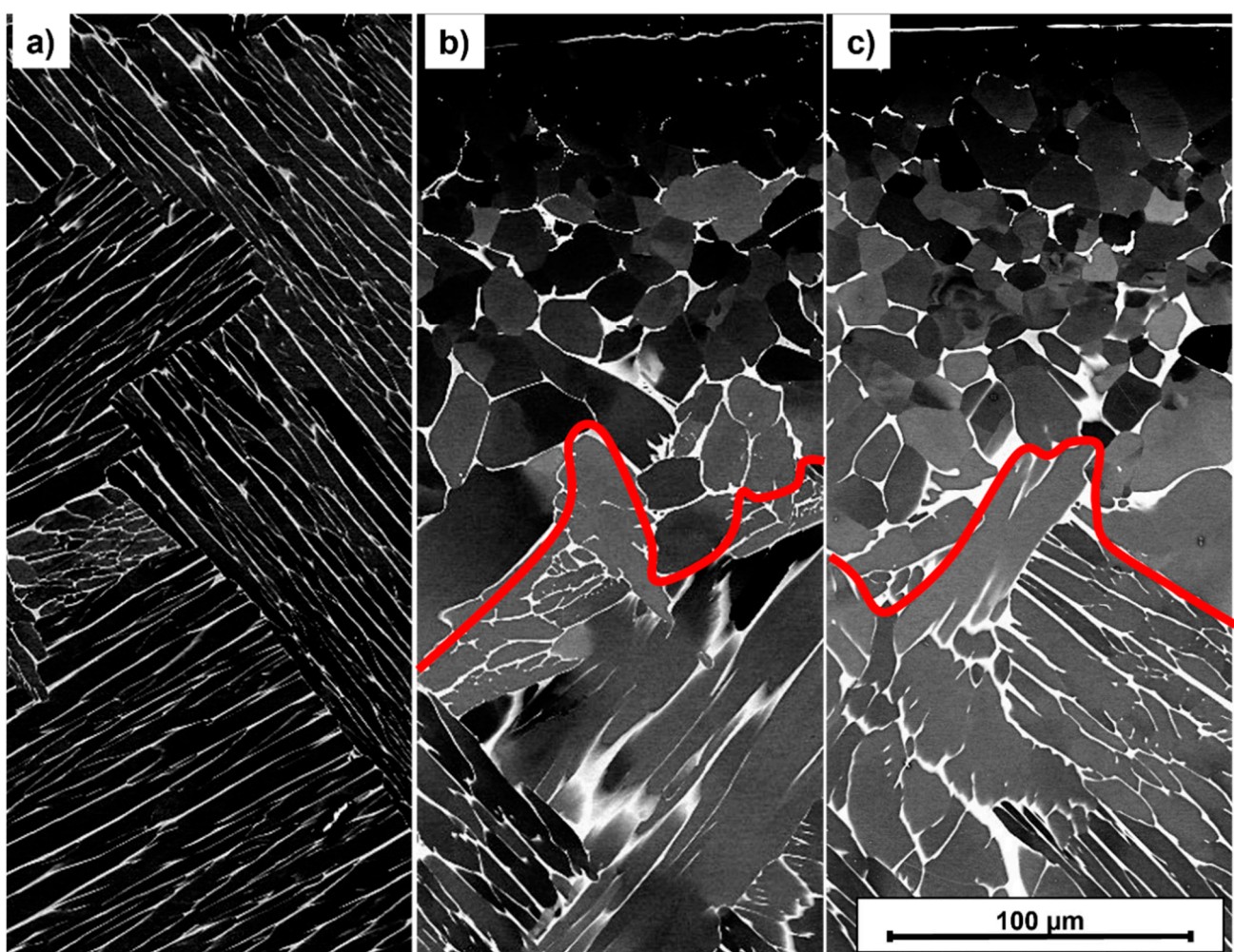

**Figure 5.** SEM-BSE Micrographs of the near-surface area (sample's surface is located on top) after solution treatment A (**a**) and THT acc. to path A (**b**) and path B (**c**); the red line indicates the boundary between equiaxed and lamellar grain morphology.

The dark areas in Figure 5 correspond to the $\alpha$ phase, while the light areas represent the $\beta$ phase. The THT microstructures in Figure 5 show an equiaxed morphology near the surface (above the red line) and a lamellar one towards the sample center (below). The microstructural states differ quantitatively in terms of the distance from the surface to the red line (penetration depth) and $\alpha$ particle size, as quantified in Table 6.

**Table 6.** Metallographic data of the microstructure after solution treatment and dehydrogenation.

| Condition | Penetration Depth | $\alpha$ Particle Size/Lamella Width ($\mu$m) per Distance Range to Surface | | | |
|---|---|---|---|---|---|
| | ($\mu$m) | 0–120 $\mu$m | 140–280 $\mu$m | 280–420 $\mu$m | Center |
| path A | $160 \pm 25$ | $17 \pm 3$ | $19 \pm 1$ | $14 \pm 2$ | $14 \pm 1$ |
| path B | $172 \pm 29$ | $23 \pm 2$ | $19 \pm 3$ | $14 \pm 1$ | $16 \pm 1$ |

The larger penetration depth is measured in the path B sample, the smaller $\alpha$ particle size in the near-surface area in the path A sample (Table 6). In the areas far from the surface (140–420 $\mu$m distance to surface and at center), no significant difference in $\alpha$ particle size

can be detected between the two THT paths, although the path B sample reveals a larger particle size in the center. This can be explained by the fact that the significantly longer hydrogenation time of the path B sample results in a larger driving force of grain growth during hydrogenation. The investigated THT processes thus induce a microstructure gradient. However, it must be verified in further investigations whether and to what extent this improves the mechanical properties. The corresponding phase analysis after THT is depicted in Figure 6 in terms of XRD diffractograms.

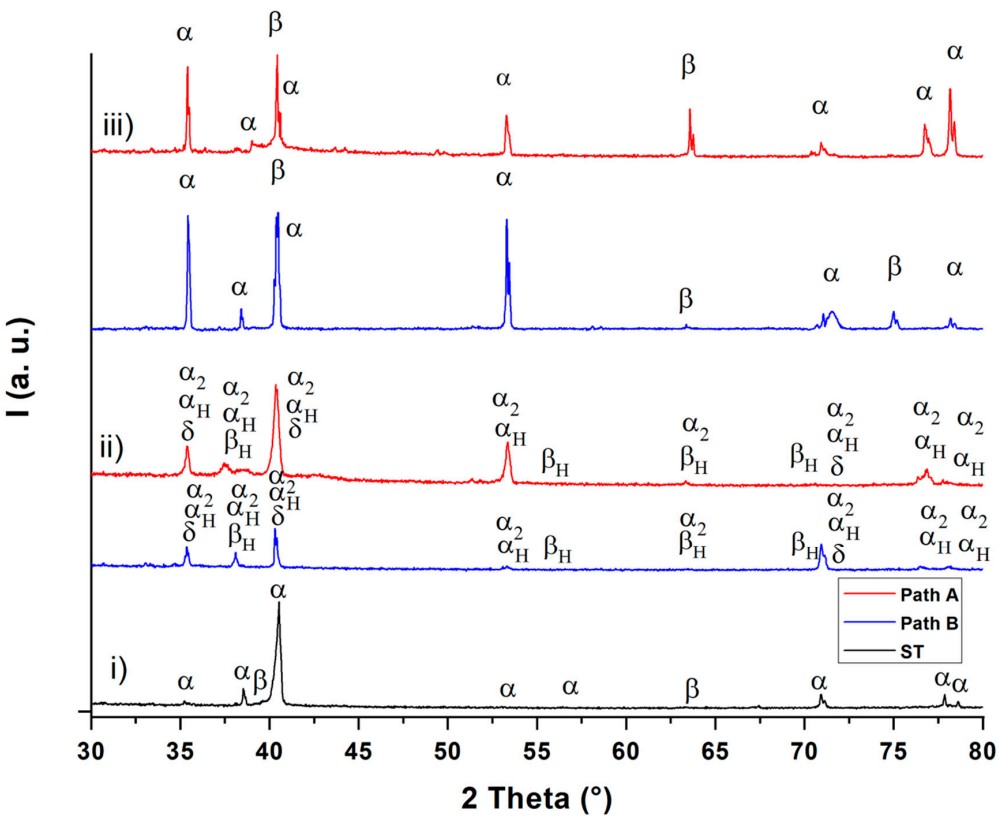

**Figure 6.** XRD diffractograms after solution treatment (ST) (**i**), hydrogenation (**ii**) and dehydrogenation (**iii**).

The XRD diffractograms (Figure 6) show the existence of the $\alpha$ and $\beta$ phases after solution treatment (diffractogram (i)) and after dehydrogenation (diffractogram (iii)). After hydrogenation, the $\alpha_2$ (Ti$_3$Al) phase (hexagonal D019 structure) and the $\delta$ (TiH2) phase (fcc) are observable in addition to the hydrogen-enriched $\alpha$ and $\beta$ phases ($\alpha_H$ and $\beta_H$) for both THT paths (diffractogram (ii)). These observations are in good agreement with the results from Liu et al. [17]. These observations can be applied to the micrographs in Figure 2 and, according to [18], suggest that the dark gray areas correspond to the $\alpha$, $\alpha_2$ and $\alpha_p$ phase, while the white areas represent the $\beta$ phase and the $\delta$ phase. The XRD diffractograms of the two THT routes differ only slightly in both the hydrogenated and dehydrogenated condition. Since the obtained hydrogen concentration after hydrogenation is, according to the Ti-6Al-4V/H phase diagram [4] (see Figure 1), not sufficient to cause $\delta$ phase formation during the isothermal hydrogenation step, this phase probably forms via eutectoid reaction during cooling from the hydrogenation temperature. The $\alpha_2$ phase is formed because of the $\beta$-stabilizing effect of hydrogen, which leads to an enrichment of aluminum in the $\alpha$ phase. According to Froes et al. [19], $\alpha_2$ is formed when the Al concentration at the $\alpha$ phase exceeds 8 at.%. After dehydrogenation, the $\delta$ phase and the $\alpha_2$ phase are no longer detectable. This supports the hypothesis that the size of $\alpha$ particles decreases due to the dissolution of finely dispersed hydrides ($\delta$ phase) during dehydrogenation. Moreover, the dehydrogenation temperature and duration appear to be sufficient to dissolute the brittle $\alpha_2$ phase. This observation can be explained by assuming that the $\alpha$ phase, which was formed from the

β phase and therefore is poor of Al. To compensate for the Al concentration differences between the α phase particles that were not transformed from β phase, this α phase absorbs Al from the environment [4,19]. Nevertheless, further experiments for the exact phase detection and the qualitative determination of the corresponding phase volume fractions are to be done and planed.

## 4. Conclusions

The study presented intended to identify the process parameters of a THT process of Ti-6Al-4V, which allow the induction of a microstructural gradient. In order to determine a solution-treated condition suitable for a subsequent temporary hydrogen alloying, first of all, the threshold stress intensity range for fatigue crack propagation and the fracture toughness for three different solution treatments were determined and compared to each other. The lamellar microstructure turned out to show a reasonable balance of cyclic strength and monotonic ductility in combination with a high resistance to hydrogen-induced surface crack formation. Based on this, different microstructural gradients were established by annealing in hydrogen atmosphere and subsequent hydrogen degassing in vacuum. The resulting microstructure gradients were evaluated by means of SEM images, simulated hydrogen concentration distributions, hardness profile measurements, thorough metallographic examination of the microstructure and X-ray diffraction analysis of the phases formed. It was found that it is basically possible to create a microstructural gradient with a penetration depth of 160 μm (path A) to 170 μm (path B) that promises increased fatigue life as compared to a homogeneous microstructure. Compared to the reference (solution treatment condition) THT increases the hardness of the samples at 100 μm distance to surface by 13% (path A) and 21% (path B) and at 200 μm distance to surface by 3% (path A) and 5% (path B). The desired microstructural gradient was realized by two newly designed hydrogen treatment strategies that comprise a hydrogen uptake at 500 °C and hydrogen partial pressures of either 100 mbar or 20 mbar and a subsequent hydrogen degassing at 750 °C. The future work will focus on the quantitative determination of the resulting phase fractions by means of TEM as well as on the change in fatigue life at low and high numbers of cycles (LCF and VHCF, respectively). For this purpose, respective fatigue tests will be carried out on Ti-6Al-4V samples after standard heat treatment and the two developed thermohydrogen treatments, which need to be supplemented by an aging treatment.

**Author Contributions:** Conceptualization, H.-J.C. and C.D.S.; methodology, C.D.S.; software, C.D.S.; validation, H.-J.C., A.V.H. and C.D.S.; formal analysis, H.-J.C. and C.D.S.; investigation, C.D.S.; resources, C.D.S.; data curation, C.D.S.; writing—original draft preparation, C.D.S.; writing—review and editing, H.-J.C. and A.V.H.; visualization, C.D.S.; supervision, H.-J.C.; project administration, H.-J.C.; funding acquisition, H.-J.C. and C.D.S. All authors have read and agreed to the published version of the manuscript.

**Funding:** This research was funded by the DEUTSCHE FORSCHUNGSGEMEINSCHAFT, Bonn, Germany (project 470236376) and a scholarship of the STIFTUNG DER DEUTSCHEN WIRTSCHSFT (SDW), Berlin, Germany.

**Institutional Review Board Statement:** Not applicable.

**Informed Consent Statement:** Not applicable.

**Data Availability Statement:** The data that support the finding of this study are available from the corresponding author upon reasonable request.

**Acknowledgments:** Special thanks go to OTTO FUCHS KG, MACDERMID ENTHONE GMBH and ATOTECH GERMANY GMBH for providing samples and consumables free of charge plus the Micro- and Nanoanalytics Facility Siegen (MNaF) for supporting the characterization of materials. We acknowledge support in experimental implementation by Seyma Aslan, Julian Müller and Simon Ortmann.

**Conflicts of Interest:** The authors declare no conflict of interest.

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
