# Peer review of "Hydrogen as a Temporary Alloying Element for Establishing Specific Microstructural Gradients in Ti-6Al-4V"

_metals, doi:10.3390/met12081267_

Round 1
Reviewer 1 Report
A brief summary : This study is concerned with controlling the microstructure of Ti-6Al-4V alloy by applying the THT (thermo-hydrogen treatment) technique.
General comments :The manuscript has been written in a slipshod style, so that the present reviewer has found it difficult to read thorough until the end. It is below the level of articles in international scientific journals. The authors are required to examine word by word, sentence by sentence, paragraph by paragraph, section by section and reference by reference with a special emphasis on the logic and technical implications.
Specific comments for examples : 1) The abstract is redundant. The first 8 lines are diffuse and verbose. What are the main results and technical implications of the present research? 2) The authors cite their previous publications [14,15]. They were published in local and inaccessible conference proceedings. Furthermore, one does not know if the publications had passed critical reviewing.The authors are requested to describe their research in a self-consistent manner. 3) In the line 35th of the text, many references are cited as a group of [4-12]. The references are not arranged in chronological order at the References. Is there any special meaning? 4) Concerning Figure 1, what is the meaning of the dotted bold straight line between 0 and A? In the caption, what does Ti-64/H denote? Ti-6Al-4V alloy? 5) In the line 106th, paths A and B are introduced. It is requested to display the drawings of the heat patterns, preferably referring to Fig. 1.
These are just examples. It would be necessary to examine the whole manuscript carefully to be welcomed by the Metals community in the world.
Author Response
Thanks for the remarks. I will answer to the specific comments step by step
1) a) The attached file shows a rephrase of some parts of the abstract. It includes the motivation, a brief literature background, the methods used and the main results that can be presented so far. Since this is the common sense of useful information that should contain an abstract can you please be specific which information exactly can be removed?
- b) Is this answering your question "What are the main results and technical implications of the present research?"?: The study shows that hydrogenation at 500°C and dehydrogenation at 750°C enables the generation of a promising microstructural gradient.
2) The author's previous publications that are cited are the reference 14 and 16, Ref. 15 is a work from Berg and Wagner. Both, ref. 14 and 16, are critically, peer reviewed paper published in proceedings on established international and nation conferences with a high reputation (international: titanium world conference, national: Tagung Werkstoffprüfung). Furthermore the paper are accesible:
[16]https://doi.org/10.1051/matecconf/202032112017
3) The references are summed up to give an overview what kind of work hat been done in the field. At the further paper they are cited so their order is given by their appearance in the text. Only ref. 10-12 are not mentioned again in the text. They proof that THT was successfully applied to various Ti-alloys. Now they are ordered chronologically.
4)You find the answer to the first question in lines 37-38 (line shows the temperature and the resulting H content after the first step, the solution treatment, which is carried out in air).
5) In the attached file figure 1 takes this remark into account.

Reviewer 2 Report
Figure 3 give calculated hydrogen concentration profiles according to PathAand path B. suggest that: it should be described the essential difference between path A and path B more detaily.
Author Response
Thanks for the comment. I replaced figure 1 with a more detailed description.

Reviewer 3 Report
Line 64: what do you mean by 'improves the properties relevant to the application'. Which properties? what kind of improvement?
Line 74: what do you mean by' Hence surface sites of a component can be reached which are mechanically difficult to access.' The surface of the alloy is difficult to reach ny mechanical processin? This is really not obvious to me.
Line 75-76: So, do you mean that because THT involves gaz then it could reach any surface? To me this is obvious.
Lines 77 to 80: This sentence is too long and not clear.
Lines 80- to 85: This sentence means that the oblect of the investigation was to select hydrogenation and dehydrogenation parameters. Is it the case?
Line 98: 'Executed' ???
Lines 130 to 132: You determined the gehyrogenation and hydrogenation temperature by SEM !!! Could you explain? You presented only one micrograph on a sample at 500C.
Section 3.1: Sorry but I do not understand anything here. I may be stupid but it is not clear how the values of table 2 were measured. Also, what are the error bars on these values? The number of significant digits displayed seems arbitrary to me.
Figure 2: you rpesent a lamellar and a bimodal structures. These structures are from which alloy? Why are they differents? I am totally lost here and now have no idea what you are talking about.
I stop reviewing here. This paper is too confused to me.
Author Response
Thanks for the remarks. I will go through them step by step:
line 64: The new version of the manuscript adds "strength at cyclic loading in HCF and LCF regime".
Line 74: surface sites of a component with a complex geometry
Line 75-76:
The study presented compares the fatigue crack propagation resistance and the fracture toughness. They are displayed as a function of three different solution treatment conditions. The comparison is used to select suitable solution treatment parameters. The solution treatment must evoke a morphology that is suitable for the surface-far area of the final microstructural gradient [15].
ll 80-85: yes
l 98: heat treated
ll 130-132: used for a qualitative evaluation of the stereological parameters
see also ll 148-149
3.1 The description is here (ll. 90-99):
the solution treatment conditions after [14] were assessed with respect to the threshold stress intensity range for long cracks (ΔK0) and the fracture toughness (KIc or KQ). ΔK0 was determined with four-point bending fatigue tests at 100 Hz (stress ratio: 0.1) by using the load-shedding method, in which the stress amplitude is reduced according to an exponentially decreasing function. ΔK0 is achieved when the crack propagation rate da/dN is less than 10-11 m/load cycle. For the determination of fracture toughness compact tension tests were carried out at a servo-hydraulic testing machine using a strain gauge. The specimen geometries were designed according to the guidelines of ASTM E 647 (ΔK0) and ASTM E 399 (KIc or KQ).
THe number of digits is replaced. The display of the result is given as it is used in the literature, without error bars.
Figure 2: New subscription, and new headline at table 1:
Solution treatment |
A: lamellar |
B: bimodal |
C: equiaxed |
. Exemplary micrographs of the near-surface area of Ti-6Al-4V samples after solution treatment A (a) and B (b), hydrogenated at 500°C for 2 h.
"I stop reviewing here. This paper is too confused to me."
I am sorry for that. If you can give me specific remarks like you did untill line 132 I am sure I am able to answer them.

Round 2
Reviewer 1 Report
The revised manuscript is significantly improved, but is still below the level acceptable for scientific papers in international journals.
1. The text relies heavily on references [15]. The reference is succinct and written in German. Non-Germans usually have no access. Authors are requested to provide data on hydrogen solubility and hydrogen diffusion coefficient and associated references in this manuscript (lines 146-148).
2. The logical structure of the manuscript needs to be improved.
2-1 Figure 1 is a a phase diagram mixed with Path A and Path B. A citation of the phase diagram might be essential in the Introduction. However, the mixture of diagrams for Path A and Path B are confusing and difficult to be understood. The THT process might be illustrated in the section of Materials and Method separately for Path A and Path B. By the way, the present reviewer does not agree to use the word "Path". Instead, "Plan A" or "Scheme A" seems to be more appropriate.
2-2 The Materials and Methods section is recommended to subdivide into three. For example, 2-1 materials preparation and THT treatment, 2-2 testing method and 2-3 numerical simulation. The 2-1 section could include the above-mentioned THT patterns for A and B separately.
3. If the only conclusion drawn from this study is expressed in lines 343-345, the significance of that statement should be clarified in terms of microstructural gradients and mechanical properties.
4. What are the advantages and disadvantages of A and B?
5. As the authors have revised the text as suggested, the abstract will be changed to a more concise and concise style.
Author Response
Thanks again for the remarks. I will answer to the specific comments step by step:
- I added Table 2 and 3, which is an excerpt of the results from [15]
- „A citation of the phase diagram might be essential in the Introduction“: I cited the diagram in the former version of the manuscript and I added a part to ll. 36-37. Moreover, the addition of path A and B into the diagram was a remark of one of the other reviewer. The expression path is still a better choice since it is not a plan or a scheme but both paths will be followed and their resulting fatigue properties will be evaluated and compared to each other.
- Done
- see ll: 352-357
- changed: see ll 117-120
- ok

Reviewer 3 Report
I still find the style of writing confusing but for a specialist it may be OK. Below are some editorial comments
Line 9 : What are ‘technical components’?
Line 82: What is a ‘complex-shaped application’?
Line 291, Table 4. These numbers do not make sense from the point of view of significant digits. For example, 159.78±24.91 should be 160 ± 25. In the same way16.97±2.73 should be 17±3. This is elementary knowledge.
Figure 6 and lines 305 to 326: This has to be completely revised. The indexation of the X-ray patterns is very strange. The phases alpha, alpha2 and delta are indexed to the same peaks! How is it possible? Also, the delta phase is fcc and not bcc!!! All this indexation seems to be wishful thinking and not based on the actual patterns of the phases that are supposed to be there. A new analysis on solid crystallographic arguments is needed.
Author Response
Line 9 : changed
Line 82: changed
Line 291, Table 4: changed
Figure 6 and lines 305 to 326:
l 312 changed bcc to fcc
The results oft he XRD scan show at one peak (e. g. 40.5°) three different finer peaks at a high intensity. Each of these peaks can be assigned to one phase. This is also in consonance with the results of [19].

Round 3
Reviewer 1 Report
Nothing.
Author Response
Thank you for the fruitful discussion. Some further changes have been applied, which you can see in the newest version of the paper.
Reviewer 3 Report
Line 9: Components for technical applications means nothing! This is too general. A component could be anything! Same with 'technical applicatioons', it could be anything. Please be more precise about what you are talking about.
Line 82: This sentence is still unreadable. It makes no sense from a grammatical point of view.
Table 6 has not be changed. The number 159.8±24.9 is still there as well as the others. In my previous comment I expalined that this does not make sense in term of significant digits . This number should be written 160 ± 25. In the same way16.97±2.73 should be 17±3.
Lines 320 to 342: How could you assign one Bragg peak to 3 different phases! You wrote in line 326 that the dark grey areas correspond to the α, α2 and αp phase. How could you do that! This does not make any sense. You should do a Rietveld refinement in order to quantify each phase. Figure 6 and the discussion that follw are unacceptable and the paper should be rejected for this reason.
Author Response
Thank you for the fruitful discussion.
Line 9: changed to
Parts of vehicles like landing gear components of aircrafts,
Line 82: changed to
Hence THT enables a local microstructure adaptation of complex geometries, like tubes with a variable wall thickness, that cannot be surface hardened via thermomechanical surface treatments
Table 6: changed
Lines 320 to 342:
The results are measured like figure 6 states it and they are discussed in relation to findings in the literature [4,17,18]. These are papers published in highly reputable journals. I checked the data of the XRD scan and changed figure 6. Moreover I added the sentence
“Nevertheless, further experiments for the exact phase detection and the qualitative determination of the corresponding phase volume fractions are to be done and planed.“ ll 343-345 and
„ The future work will focus on the quantitative determination of the resulting phase fractions by means of TEM as well as…“ in ll 366-367
Since TEM experiments for the validation of the resulting phases and phase fractions are planned this is mentioned in the outlook. But since [4,17,18] hydrogenated up to a comparable H concentration value these phases are also expectable in the samples of this study. This is not a new finding but the XRD measurements, which deliver reasonable results, are just used as a validation that the mechanisms of the microstructural evolution described in [4,17 and 18]. Therefore, I do not agree that the discussion of this figure is unacceptable since it takes the literature into account. But I do agree that there are further measurements to be done. The paper provides an insight into the current status of the project, like it was presented on the LightMAT 2021. Further publications are in progress and will give more detailed results in terms of the phase detection.